# CRYSTALSEG: AUTOMATING SYNCHROTRON TOMOGRAPHIC RECONSTRUCTION SEGMENTATION FOR CRYSTALLOGRAPHY WITH PHYSICALLY GUIDED SIMULATION

## ABSTRACT

Automated 3D segmentation of tomographic volumes is a critical bottleneck in long-wavelength X-ray crystallography, a technique crucial for drug development and validating structural models from systems like AlphaFold3. This segmentation is a prerequisite for ray-tracing absorption correction, which is necessary for data processing in X-ray crystallography experiments. However, it is currently performed manually by experts, which is a process that is slow, costly, and prevents full automation of the scientific pipeline. The primary barrier to automation is the prohibitive expense and difficulty of collecting annotated segmentation data. To address this data scarcity problem, we present **CrystalSeg**, a novel, GPU-accelerated simulation and segmentation pipeline. It generates vast amounts of annotated data by simulating synchrotron X-ray tomography images and their corresponding reconstructed 3D volumes. We demonstrate that segmentation networks trained on CrystalSeg's synthetic data achieve dramatic performance gains over models trained on limited real data, with **improvements of 29.2% in Recall, 30.5% in IoU, and 24.9% in F1 score** for finding the crystal. CrystalSeg effectively reduces the expert labor required for segmentation from hours to minutes. More importantly, it enables, for the first time, a fully automated solution for ray-tracing absorption correction in long-wavelength crystallography, making this advanced structural biology technique more scalable and accessible.

## 1 INTRODUCTION

Long-wavelength X-ray crystallography plays a crucial role in experimentally determining protein structures, and localizing and identifying target atoms through anomalous scattering (El Omari et al., 2024). AlphaFold3 (AF3) (Abramson et al., 2024) can provide excellent geometric priors of the protein structure but cannot measure the identity of target atoms, or the properties of the redox state inside the protein. Combining AF3 with X-ray crystallography data allows for validating the predictions from AF3, designing fragment-based drugs (Ma et al., 2024), and solving the structure of previously unseen proteins.

However, this powerful combination is hindered by a critical data processing bottleneck: **ray-tracing absorption correction**. This physical correction step is mandatory for processing long-wavelength data, as the crystal sample itself can introduce non-linear errors in the measured X-

Figure 1: A typical sample's tomography reconstruction (top) and its segmentation (bottom). The segmentation highlights the crystal (in light purple), the surrounding mother liquor (in semi-transparent black), and the mounting loop (in coral red) (Kazantsev et al., 2021). Best viewed in colour.

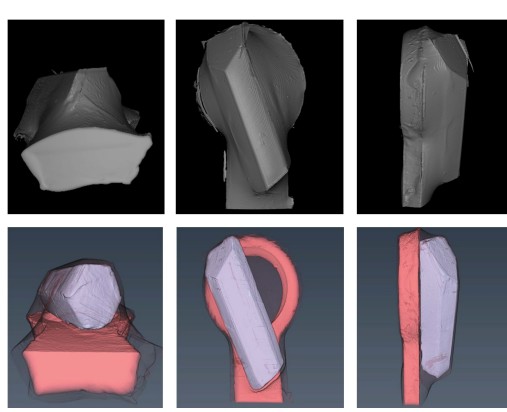

Figure 2: Illustration of how synchrotron tomography segmentation helps long-wavelength X-ray crystallography to validate and correct the predictive models from AlphaFold3 (Abramson et al., 2024). By using model-based segmentation, the whole data processing can be automated and the runtime for data annotation can be reduced from 4+ hours to around 10 seconds. The visual image, tomography reconstruction, and segmentation of the crystal sample are shown (Kazantsev et al., 2021).

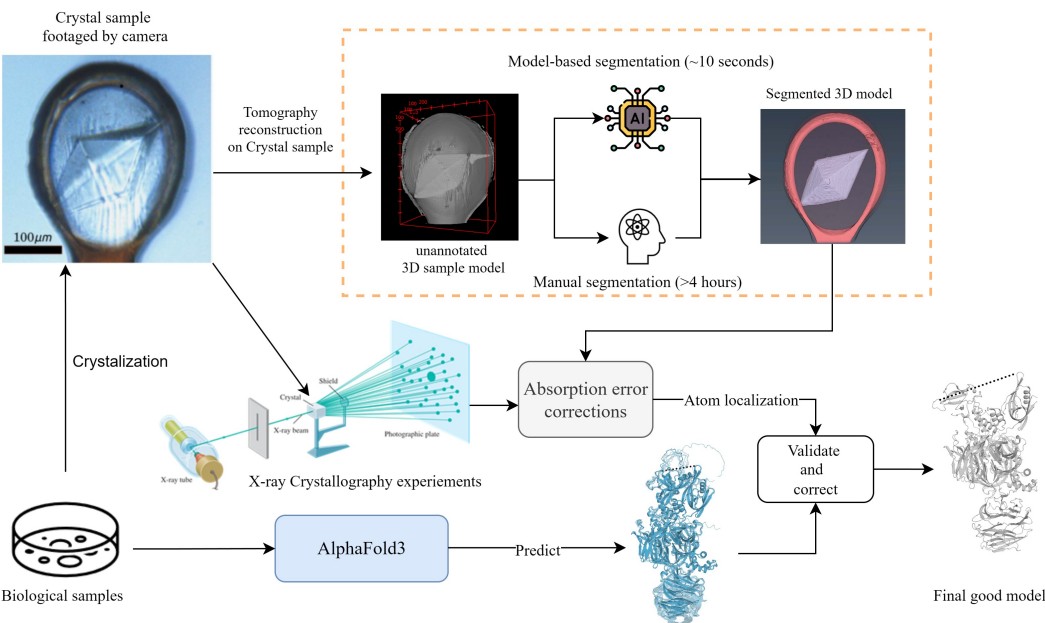

ray intensities (as shown in Figure 2, further details in Section §S1). Accurately performing this correction requires a precisely annotated 3D model of the sample, which is typically acquired via X-ray synchrotron tomography. The problem is that this tomographic volume must be **manually segmented** to label the crystal, its surrounding solution (mother liquor), and the mounting loop (Figure 1) (Lu et al., 2024b;a). This manual, multi-class segmentation task is extremely time-consuming, which often **takes hours per sample**, and prevents the entire scientific pipeline from being fully automated.

Intuitively, this 3D segmentation task can be automated with standard deep learning techniques based on recent progress in other fields for 3D segmentation (Isensee et al., 2021; Yu et al., 2024; Azad et al., 2024; Lin et al., 2022). However, this is highly challenging due to significant data-related hurdles. Protein crystal samples are often expensive and delicate, making the collection and annotation of large-scale, high-quality training datasets practically infeasible. Furthermore, the samples exhibit high variance in their physical properties (e.g., absorption and phase contrast), leading to diverse intensity distributions in the reconstructed volumes. This problem is compounded by inconsistent sample positioning and varied morphologies. Unlike segmentation in medical imaging, which often targets well-defined organs, segmentation in crystallography must handle a wide-ranging, sparse, and variable domain, making robust generalization a significant challenge. For example, the crystal in Figure 1 is longitudinal and at the top of the loop, while that in Figure 2 has a diamond-like shape and is in the middle of the loop. These challenges make automatic segmentation significantly more difficult.

To overcome the above data scarcity problem, in this paper, we introduce CrystalSeg, an innovative method to simulate synchrotron tomography reconstructions with physical guidance, for generating synthetic training data to train a segmenter. Physical guidance by simulating X-ray wave propagation can help ensure that the synthetic data accurately reflects the interaction between X-rays and the material properties of the sample, including factors such as absorption, scattering, and phase shifts. This guidance introduces a level of realism that allows DNNs trained on synthetic data to generalize better to segment real-world synchrotron tomography datasets. We demonstrate that our

simulation method can be accelerated by NVIDIA GPUs. By incorporating hyperparameters, such as refractive indices of the materials, and randomising the positions of the crystal, CrystalSeg can provide high-quality, high-volume, synthetic data for training Deep Neural Networks (DNNs) to achieve automatic synchrotron tomography segmentation.

Our results demonstrate that our method not only gains accurate simulated outcomes but also significantly improves DNN training efficiency. CrystalSeg is **the first fully automated approach** for efficiently generating large volumes of annotated training data, enabling the training of a 3D segmentation model for synchrotron tomography reconstruction data. This advancement facilitates the first fully automated ray-racing absorption correction in long-wavelength crystallography. Tasks that once required over four hours of manual intervention and annotation can now be completed in only seconds through automated segmentation.

The main contributions of this paper are as follows:

- We introduce the first fully automated solution for ray-racing absorption error correction in long-wavelength crystallography with an automatically annotated 3D model of the crystal sample. This method enhances efficiency and accessibility for users.

- To address the challenges of training a DNN for automated 3D model annotation, we propose CrystalSeg, an innovative and efficient method for simulating synchrotron tomography reconstructions, accelerated by NVIDIA GPUs for improved speed.

- Our approach demonstrates accurate simulation results, achieving high SSIM and PSNR values that closely match those from real experimental data.

## 2 RELATED WORK

### 2.1 TOMOGRAPHY RECONSTRUCTION SEGMENTATION IN SYNCHROTRONS

Synchrotron X-ray tomography is a well-established technique, supported by numerous dedicated beamlines worldwide that provide high-resolution imaging for a range of applications. When applied to delicate and expensive samples, such as protein crystals, synchrotron tomography reconstruction reveals challenges with limited annotated data to train a deep neural network (DNN). The composition and structure of these samples can lead to varying levels of absorption and phase contrast, influenced by differences in sample size, shape, and material properties. Furthermore, experimental artefacts, such as beam hardening and ring artefacts, can lead to noisy reconstruction results. Also, variations in the crystal's position relative to the surrounding material often require case-specific adjustments for accurate analysis. These factors make it challenging to develop robust, generalized models, especially in crystallography.

Semi-automatic segmentation techniques can assist in synchrotron tomography experiments. Such techniques include intensity thresholding, which separates regions based on differences in intensity (Alvarenga de Moura Meneses et al., 2011); region growing, which expands a region from a "seed" point to include adjacent points with similar intensities (Kazantsev et al., 2021); and topological watershed, which separates regions based on gradient differences at the edges (Kornilov et al., 2022). Although these methods can effectively differentiate between regions, they still require human intervention to assign the correct material to each segmented region.

Numerous DNN methods have been developed for segmentation tasks in synchrotron X-ray tomography. These methods typically involve training DNNs on annotated real data. Some approaches focus on segmenting specific materials or biological molecules that exhibit similar absorption contrasts or shapes after reconstruction Torbati-Sarraf et al. (2021); Davydzenka et al. (2022); Yang et al. (2021). Alternatively, some methods use publicly available simulation data or simulate a single material type to train segmentation networks that distinguish between the foreground and background regions Moebel et al. (2021); Lin et al. (2022). In synchrotron tomography segmentation, particularly in crystallography, segmenting components such as the mother liquor, which often shares absorption characteristics similar to those of the crystal, presents unique challenges. The relative positions of the mother liquor, crystal, and mounting loop can vary significantly between samples, complicating segmentation. Furthermore, a primary limitation of DNN-based segmentation on real data is the extensive effort required for data collection and annotation.

## 2.2 Tomography data simulation

Generative Adversarial Networks (GANs) and diffusion models have been applied to synthetic medical CT generation (Yu et al., 2024; Friedrich et al., 2024). However, these models usually require hundreds of diverse, well-annotated cases, while labeled synchrotron datasets are scarce. Synchrotron tomography also exhibits substantial variability in crystal shape, size, orientation, and position. Training a GAN or diffusion model under this variability is therefore highly challenging. Moreover, most available open-source pre-trained weights are trained on medical images, creating a domain gap that further limits transferability. In contrast, simulation by physical guidance offers an alternative by utilizing experimental equipment parameters, such as the X-ray source and detector type, along with the properties of the experimental sample. By alternating the refractive indices and designing 3D synthetic models of the crystal sample by CAD software, a large amount of synthetic data with physical guidance can be generated.

A more accurate approach is to generate synthetic tomography images via physics-based simulation imaging (Ching & Gürsoy, 2017; Kazantsev et al., 2018; Faragó et al., 2017; Unberath et al., 2018; Gopalakrishnan & Golland, 2022). A significant limitation of current tomography simulations by physical guidance is their reliance on physical models that are either overly simplified or computationally prohibitive for large-scale data generation. Many simulators, particularly phantom-based tools, treat objects as generic attenuation maps with simple geometry and pure absorption contrast (Ching & Gürsoy, 2017; Kazantsev et al., 2018). While recent differentiable renderers achieve high anatomical realism, they must also simplify physics for tractability, focusing on absorption and overlooking the crucial phase-contrast phenomena that arise from wave propagation (Unberath et al., 2018; Gopalakrishnan & Golland, 2022). This abstraction fails to generate the high-fidelity edge enhancement and material-aware contrast governed by the precise, energy-dependent refractive indices of constituent materials. Conversely, simulators that do model these complex wave-based interactions are often too computationally expensive for deep learning frameworks. Their design as high-fidelity physics workbenches requires an intricate, manual setup of source and detector parameters for each simulation, which is impractical for programmatically generating the large, diverse datasets required to train robust models (Faragó et al., 2017). This forces a critical trade-off, leading to a simulation-to-reality gap that directly impacts model performance. A network that has been trained on simplified projections will fail to generalize to real-world experimental data, as it has never been exposed to the crucial, material-dependent phase effects that are paramount in applications like high-resolution crystallography.

## 3 Methodology

### 3.1 Simulation pipeline

As illustrated in Figure 3, the overall simulation process consists of 1. designing 3D synthetic models of the crystal sample using CAD software, 2. simulating projection images by simulating X-rays propagating through the crystal sample and finally reaching the detector, 3. rotation of the 3D synthetic models over 180°, 4. performing tomography reconstruction on that series of projection images using **filtered back-projection (FBP)** using the *TomoPy* (Pelt et al., 2016) software.

The synthetic crystal samples are not manually modeled but are generated within a CAD software *Blender* via its Python API, which is an automated and high-throughput pipeline. This process is grounded in physical principles: each crystal's morphology is constructed from its crystallographic data, including its crystal system, point group symmetry operations, and Miller indices (hkl) for its characteristic faces. Similarly, the mounting loop is constructed based on realistic dimensions to ensure consistency. The mother liquor is simulated using *Blender*'s integrated fluid physics engine, where a fluid domain is established around the crystal and loop. The simulation output from *Blender* is in *Mesh* format, and the 3D volume of a tomography reconstruction is in *Array* format. Hence, the simulation dataset from *Blender* is voxelized using *Open3D* (Zhou et al., 2018). The details of simulating projection images are described in Section 3.2. After simulating the wave propagation between the sample and the virtual detector, the intensities of the wavefield are recorded as synthetic tomography projection images. By rotating the simulated sample over 180° with 0.2° as an increment, there is a total of 900 synthetic tomography projection images in a dataset, as shown in Figure

Figure 3: The simulation process of CrystalSeg

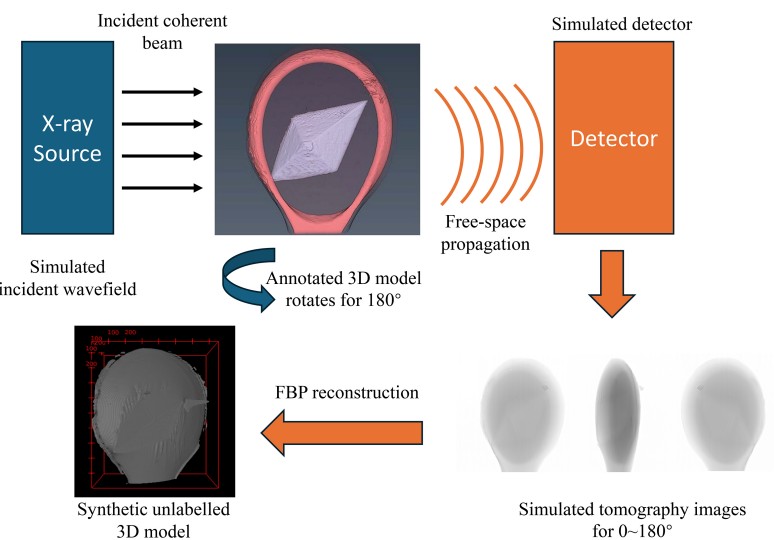

3. Then, using FBP to reconstruct the series of 900 projection images to get the reconstructed 3D volume.

## 3.2 SIMULATING X-RAY PROJECTION IMAGES BY WAVE PROPAGATION PRINCIPLE

In a real tomography reconstruction experiment, particularly at low energies (3 keV - 5 keV at long wavelength), there are significant edge effects at the boundaries between different materials. This is because the sharp interface of the crystal and the mounting loop incur significant phase contrast, not just absorption, when the previous work may ignore (Ching & Gürsoy, 2017; Unberath et al., 2018; Gopalakrishnan & Golland, 2022). To achieve a high similarity simulation dataset that captures these nuances and effectively trains deep learning models, a comprehensive understanding and simulation of the physical principles of X-ray wave propagation are essential. The overall process can be treated as an incident monochromatic wavefield, typically a plane or spherical wave, propagating through an arbitrary number of 3D objects. The wavefield's intensity is then captured at a virtual imaging plane (detector in reality) at a certain distance from the source (Born & Wolf, 2013). We utilized the multislice wave propagation technique (Kirkland, 1998) to discretize the 3D object into a stack of 2D slices to allow efficient parallel computing. This entire wavefield propagation chain, including object interaction and free-space diffraction, is efficiently computed on NVIDIA GPUs using CuPy for computational speed, which is critical for generating large datasets.

**Incident wavefield**. A point source emits a monochromatic wavefield $u_0(\mathbf{x}, z_1)$ with wavelength $\lambda$, where $\mathbf{x}$ represents the 2D coordinates perpendicular to the X-ray incident axis $z$, and $z_1$ is the distance to the first object in the beam path. The intensity distribution of the wavefield is given by $I_0(\mathbf{x}, z_1) = |u_0(\mathbf{x}, z_1)|^2$. For a spherical wave, when $z_1$ is sufficiently large (as in synchrotron common setups), the spherical phase profile is approximated to (Faragó et al., 2017):

$$u_0(\mathbf{x}, z_1) = \sqrt{I_0(\mathbf{x}, z_1)} \, e^{jkz_1}, \tag{1}$$

where $k = \frac{2\pi}{\lambda}$ is the wave number.

**Wavefield propagations**. When an X-ray beam propagates through non-vacuum objects, its intensity is attenuated, and the beam undergoes a phase shift. This behaviour is described by the 3D complex refractive index of the object $i$ at the 2D coordinate $\mathbf{x}$, located a distance $z$ from the X-ray source. The refractive index is represented as (Born & Wolf, 2013):

$$n_i(\mathbf{x}, z) = 1 - \delta_i(\mathbf{x}, z) + j\beta_i(\mathbf{x}, z), \tag{2}$$

where $\delta_i(\mathbf{x}, z)$ corresponds to the real part of the refractive index, representing the phase shift, and $\beta_i(\mathbf{x}, z)$ is the imaginary part, representing absorption within the object.

The propagation function at the exit plane of the object $i$ can be determined by integrating along the $z$-direction. This is expressed as (Born & Wolf, 2013):

$$T_i(\mathbf{x}) = \exp\left(jk \int_{z_i^-}^{z_i^+} n_i(\mathbf{x}, z)dz\right) = e^{-k(B_i(\mathbf{x})-j\varphi_i(\mathbf{x}))}, \tag{3}$$

where

$$B_i(\mathbf{x}) = \int \beta_i(\mathbf{x}, z)dz \quad \text{and} \quad \varphi_i(\mathbf{x}) = \int [1 - \delta_i(\mathbf{x}, z)]\, dz.$$

Here, $B_i(\mathbf{x})$ represents the cumulative local absorption of the X-ray as it propagates through object $i$, and $\varphi_i(\mathbf{x})$ represents the total phase shift induced by the refractive index variation. Therefore, the relationship between the wavefield $u_{i-1}(\mathbf{x}, z_i)$ at the entrance plane of the $i$-th object and $u_i(\mathbf{x}, z_i)$ at the exit plane can be described as:

$$u_i(\mathbf{x}, z_i) = T_i(\mathbf{x})u_{i-1}(\mathbf{x}, z_i). \tag{4}$$

In the case where the X-ray propagates through air or vacuum, the wavefield does not experience material attenuation but still undergoes spreading, diffraction, and phase evolution as it propagates. This free-space propagation can be modelled using the angular spectrum formalism between two parallel planes separated by a distance $\Delta z$ (Goodman, 2005). Therefore, the 2D Fourier transform of the wavefield, denoted by $\tilde{u}(\boldsymbol{\xi}) = \mathcal{F}[u(\mathbf{x})]$, describes the wavefield in terms of 2D spatial frequencies $\boldsymbol{\xi}$. The free-space propagator is given by:

$$\tilde{u}(\boldsymbol{\xi}, z + \Delta z) = \tilde{P}(\boldsymbol{\xi}, \Delta z)\tilde{u}(\boldsymbol{\xi}, z), \tag{5}$$

where the propagator $\tilde{P}(\boldsymbol{\xi}, \Delta z)$ can be written as:

$$\tilde{P}_F(\boldsymbol{\xi}, \Delta z) = \exp(jk\Delta z)\exp(-j\pi\lambda\Delta z\boldsymbol{\xi}^2). \tag{6}$$

We apply the Fresnel approximation to have this form of $\tilde{P}_F(\boldsymbol{\xi}, \Delta z)$ as X-ray illumination is parallel or weakly divergent in crystallography experiments. The Fresnel approximation is suitable when the distance between the object and the detector is large compared to the wavelength and the feature size of the object, which is common in most X-ray imaging applications. Thus, the wavefield at a distance $\Delta z$ behind the $i$-th object can be calculated using the recursive relation:

$$u_i(\mathbf{x}, z_i + \Delta z) = \mathcal{F}^{-1}\left\{\tilde{P}(\boldsymbol{\xi}, \Delta z)\mathcal{F}\left[u_{i-1}(\mathbf{x}, z_i)T_i(\mathbf{x})\right]\right\}. \tag{7}$$

In this context, the sample and detector are treated as different instances of the object $i$, enabling recursive propagation of the wavefield from the X-ray source to the detector plane.

To bridge the final gap to experimental reality, we introduce a computationally efficient detector model. A full Monte Carlo simulation is intractable for large-scale data generation. Instead, we approximate the dominant physical effects using a sequence of GPU-accelerated phenomenological models. First, the optical system's finite resolution is modeled by convolving the ideal image with a Gaussian point spread function (PSF), performed efficiently in the frequency domain (Barrett & Myers, 2013). Subsequently, the quantum nature of photon counting is introduced via a Poisson distribution for shot noise, followed by the addition of zero-mean Gaussian noise to account for electronic read noise (Hasinoff, 2014). To incorporate common instrumental artifacts, a column-wise random gain multiplier is applied, directly producing the characteristic ring patterns seen after reconstruction. This approach provides a high degree of perceptual realism and introduces the key artifacts a deep learning model must learn to be robust against.

### 3.3 Comparison with Nano Banana

We evaluated the performance of our physics-guided simulation against a state-of-the-art AI style transfer tool, Nano Banana, which is based on a pre-trained diffusion model. Such tools are valuable when limited experimental data makes it infeasible to train specialized generative models like GANs from scratch. For the evaluation, Nano Banana (Partly, 2024) was provided with five real synchrotron X-ray projection images to serve as a style guide. The target, or content image, was an ideal absorption-contrast projection image. To create a noise-free, 2D absorption-contrast image, we projected the manually segmented 3D volume by averaging its values along a single axis. The values are the distinct class labels, instead of the exact physical property. The objective was to transfer the realistic acquisition characteristics (e.g., noise, phase contrast, edge effect, intensity histogram) from the real projections onto the ideal content image. We then compared this AI-generated result with our own simulation. The precise prompts given to the AI are available in the §S2.

### 3.4 Segmentation

We segment synchrotron tomographic reconstructions using nnU-Net (Isensee et al., 2021), training both the 2D and low-resolution 3D configurations with 5-fold cross-validation (80% train / 20% validation per fold) and batch size of 3. The model selection is based on the best validation checkpoint within the best-performing fold. More details can be found in Section §S3 in the supplementary material.

## 4 Results

Figure 4: Qualitative Results of Sample A (top) and B (bottom) between real tomography projections (left) and simulated projection images (middle and right). Results from CrystalSeg are in the middle, while those from Nano Banana (Partly, 2024) are on the right.

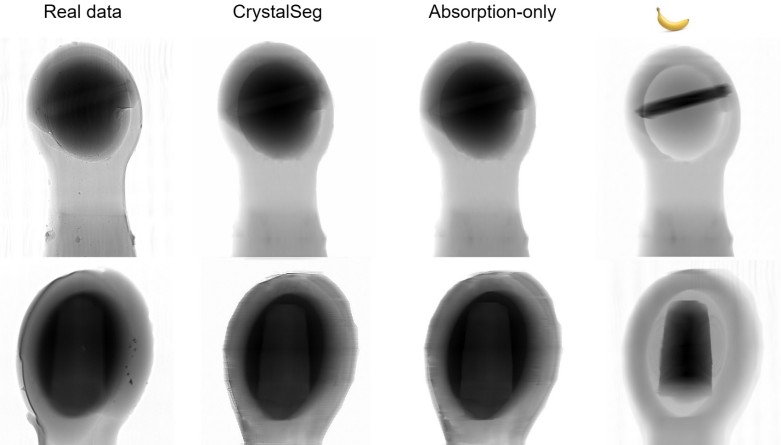

### 4.1 Simulation evaluation

To assess simulation fidelity, we start from a manually segmented 3D model derived from real data (Figure 3). The model is forward-projected to generate simulated projections, which are flat-field corrected. Agreement with measurements is quantified using SSIM and PSNR on the projections. Both simulated and real datasets are then reconstructed with FBP, and reconstructed slices are re-evaluated with the same metrics. Using $180°$ projections and the corresponding reconstructed slices, we report mean SSIM and PSNR for Samples A and B, comparing an absorption-only baseline (following Unberath et al. (2018); Gopalakrishnan & Golland (2022) with the Beer Lambert law) with CrystalSeg (Table 1).

CrystalSeg outperforms the absorption-only baseline across both samples and stages. For Sample A, projection quality is high (SSIM 0.858, PSNR 25.02 dB) and clearly exceeds baseline (0.8232,

Table 1: Mean SSIM and PSNR for Samples A and B at projection and reconstruction stages (Absorption-only baseline vs. CrystalSeg).

| Sample | Stage | Absorption-only | | CrystalSeg | |
|---|---|---|---|---|---|
| | | SSIM | PSNR | SSIM | PSNR |
| A | Projection | 0.823 | 12.86 | 0.858 | 25.02 |
| | Reconstruction | 0.895 | 21.27 | 0.9204 | 31.97 |
| B | Projection | 0.841 | 15.18 | 0.902 | 28.63 |
| | Reconstruction | 0.901 | 23.08 | 0.972 | 36.19 |

Table 2: Per-class test metrics for Crystal, Liquor, and Loop across datasets.

| Dataset | Materials | Recall | IoU | F1 Score | F2 Score | Precision |
|---|---|---|---|---|---|---|
| Only real | Crystal | 0.7134 | 0.6385 | 0.7180 | 0.7137 | 0.7406 |
| | Liquor | 0.8857 | 0.8086 | 0.8926 | 0.8881 | 0.9027 |
| | Loop | 0.7967 | 0.6720 | 0.7874 | 0.7918 | 0.7884 |
| | **Mean** | **0.7986** | **0.7063** | **0.7993** | **0.7979** | **0.8105** |
| Only syn | Crystal | 0.7997 | 0.7114 | 0.8154 | 0.8044 | 0.8473 |
| | Liquor | 0.8961 | 0.7849 | 0.8776 | 0.8886 | 0.8600 |
| | Loop | 0.8480 | 0.7334 | 0.8449 | 0.8464 | 0.8446 |
| | **Mean** | **0.8480** | **0.7432** | **0.8460** | **0.8465** | **0.8506** |
| Real + syn | Crystal | 0.9221 | 0.8332 | 0.8966 | 0.9113 | 0.8754 |
| | Liquor | 0.8966 | 0.8401 | 0.9038 | 0.8994 | 0.9114 |
| | Loop | 0.8838 | 0.8102 | 0.8864 | 0.8848 | 0.8899 |
| | **Mean** | **0.9008** | **0.8278** | **0.8956** | **0.8985** | **0.8922** |

12.86 dB), while reconstruction also improves over baseline (0.9204, 31.97 dB vs 0.895, 21.27 dB). For Sample B, projection gains are strong (0.902, 28.63 dB vs 0.8414, 15.18 dB), and reconstruction shows the largest improvement (0.972, 36.19 dB vs 0.901, 23.08 dB). Overall, CrystalSeg better matches real data, with especially large gains at reconstruction. Figure 4 reveals the critical advantages of our physics-guided simulation. Across both rows, CrystalSeg most closely reproduces the real projections: object geometry, relative attenuation, and boundary edge-enhancement are consistent. The absorption-only simulation preserves overall shape but lacks phase-contrast effects, and the edges appear softened with flattened internal contrast. The Nano Banana result maintains gross morphology but misestimates intensities as it has no physical information, such as the absorption coefficients and the refractive indices of the materials. This model-based method may also produce spatial misalignment (global shifts) of the object in the image, causing object displacement. Overall, the physics-guided CrystalSeg projections align best with the real data, matching the results in Table 1.

## 4.2 SEGMENTATION MODEL RESULTS

The significant role of synthetic datasets in training the segmentation model is highlighted in Table 2. Training with only real datasets (OnlyReal) is shown to have very poor segmentation performance. However, training with only synthetic data (OnlySyn) yields notable improvements across all mean metrics compared to using OnlyReal, achieving higher **Recall**, **IoU**, **F1 Score**, and **F2 Score**. Furthermore, the combination of real and synthetic data (All) delivers the highest performance overall, demonstrating the effectiveness of synthetic data in enhancing the model's generalization and improving segmentation quality.

Table 2 shows that augmenting with synthetic data substantially improves performance. Real+Syn achieves the best means (IoU 0.8278; F1 0.8956; Recall 0.9008), outperforming OnlySyn (IoU 0.7432; F1 0.8460) and OnlyReal (IoU 0.7063; F1 0.7993). Gains are most critical for the Crystal class: recall rises to 0.9221 (vs. 0.7997 OnlySyn; 0.7134 OnlyReal) with high F2, indicating far fewer missed crystal voxels, which can directly reduce absorption bias during the absorption correction. Liquor and loop likewise improve (e.g., liquor/loop IoU 0.840/0.810 with Real+Syn), confirming better overall material discrimination. In long-wavelength X-ray crystallography, achieving accurate voxel-wise segmentation of all material classes (mother liquor, mounting loop, and crystal) is crucial for precise absorption correction, especially for the crystal. This importance arises

Figure 5: 2D slices of the segmented 3D volumes: Crystal in Red, Mounting Loop in Yellow, and Mother Liquor in Semi-Transparent Blue. GT represents ground-truth manual labelling. Real, Syn, and Real+Syn show the segmentation models trained by those datasets.

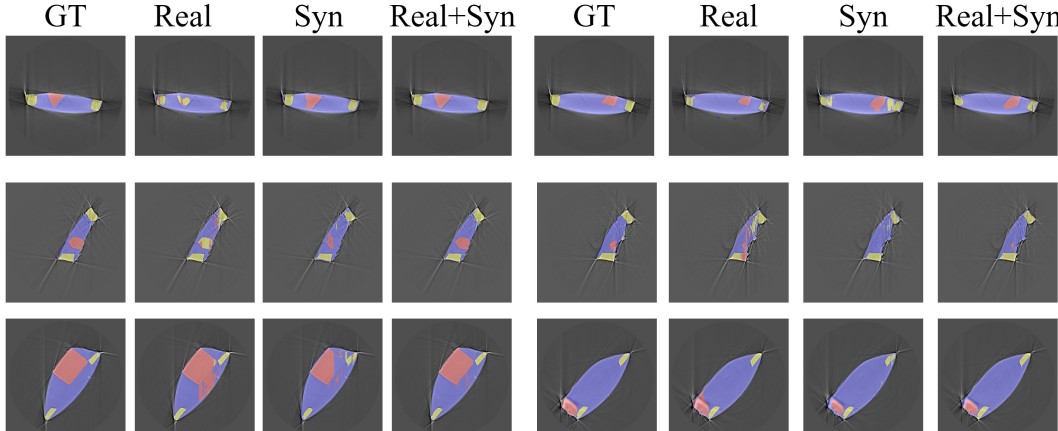

because the absorption coefficients of materials become more severe at longer X-ray wavelengths. Therefore, in addition to the overall results of segmentation, the performance in segmenting each individual material class is essential. The segmentation metrics for the crystal, mother liquor, and mounting loop are presented in Table 2. In analyzing the segmentation results for the **Crystal** class, which is the most significant class for the absorption correction process in crystallography, it is evident that the model demonstrates strong performance across several key metrics.

As shown in Figure 5, Real+Syn masks adhere closely to ground truth (GT) across diverse views: crystal (red) boundaries align with liquor (semi-transparent blue) with minimal bleed, and loop (yellow) contours remain stable and contiguous. OnlySyn already sharpens boundaries relative to OnlyReal, but occasional loop mislabelling persists. OnlyReal exhibits the weakest behaviour, with systematic under-segmentation of crystal and frequent crystal–liquor leakage, especially near loop contacts and thin edges. Together, these results show that synthetic augmentation is not merely beneficial on averages, but also it specifically optimises crystal segmentation, while improving robustness at challenging interfaces.

Two scientific case studies for comparing manual segmentation and the model trained by real and synthetic dataset are demonstrated in Section §S1 and §S4. Despite minor artefacts, the automatic segmentations produce absorption-factor histograms that largely overlap manual labels, which is the basis for ray-tracing absorption correction. Sample 1 matches with minimal precision loss, and Sample 2 shows a slight drop yet preserves key trends, so absorption statistics remain sufficient for long-wavelength crystallography while enabling a fully automated pipeline.

## 5 CONCLUSION

In this paper, we introduce CrystalSeg, a physically guided simulation method for synchrotron tomography reconstructions that provides the first fully automated solution for ray-tracing absorption error correction in long-wavelength crystallography. By leveraging physical theory and optimized computational methods to efficiently generate annotated synthetic crystal data, this approach enables the training of an automatic segmentation network, significantly reducing manual effort. As a result, CrystalSeg accelerates absorption correction in long-wavelength crystallography, supporting faster experimental validation and refinement of predictive models like AlphaFold3 (Abramson et al., 2024). Moreover, CrystalSeg can be extended to simulate additional materials relevant to synchrotron tomography, providing cost-effective training data that strengthens segmentation models across domains. By generating 3D mesh–based specimens from chemical composition (to derive attenuation/refractive indices) and incorporating domain-specific constraints (e.g., space-/point-group symmetry operations), the framework could explore a real-time synchrotron digital twin for data generation, validation, and experiment design.

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

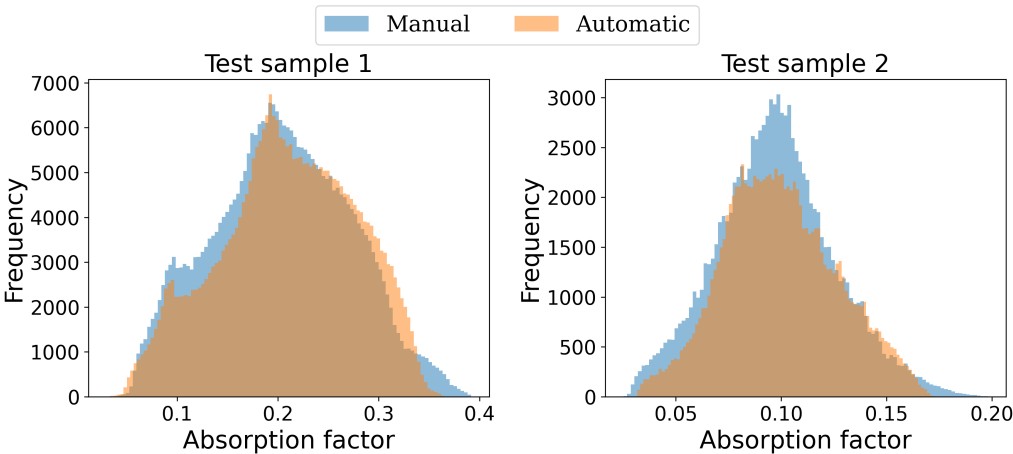

Figure S1: The histograms of absorption factors $A_{\mathbf{h}}$ between the Manual and Automatic segmentation for Test Samples 1 and 2.

## SUPPLEMENTARY MATERIALS

## S1 ANALYTICAL ABSORPTION CORRECTION

Analytical absorption correction using the ray-tracing method has proven effective in long-wavelength crystallography Lu et al. (2024b). This method calculates the absorption factor by determining the path lengths of diffracted X-ray photons, as illustrated in Figure 2, and relies on an annotated 3D model of the crystal sample. Despite we have segmentation metrics to evaluate, the segmented 3D model of the crystal samples is ultimately needed for analytical absorption correction. The equation for determining absorption factor can be expressed as:

$$A_{\mathbf{h}} = \frac{1}{V} \int_V e^{-\sum_{m=1}^{M} \mu_m L_m(x,y,z)} dV, \tag{8}$$

where $A_{\mathbf{h}}$ is the inverse absorption factor, used to correct the measured reflection intensities, $I_{\mathrm{corr}} = \frac{I_{\mathrm{meas}}}{A_{\mathbf{h}}}$. In this equation, $L_m(x,y,z)$ represents the path length that the X-ray travels through material $m$, which has an absorption coefficient $\mu_m$ for each crystal element $dV$ Albrecht (1939). The integral is evaluated numerically, and $dV$ corresponds to each crystal voxel in the segmented 3D model.

The absorption coefficients and the absorption factors are calculated by AnACor Lu et al. (2024a), and the histograms of absorption factors between manual segmentation and automatic segmentation are shown in Figure S1. Both histograms indicate a high degree of overlap, suggesting that the automatic segmentation closely replicates the distribution seen in the manual segmentation. In Test Sample 1, the agreement between the methods is particularly notable, with the frequency peaks and overall shapes closely matching, reflecting the strong performance of all three materials as shown in Table 2. Test Sample 2's histogram also shows good alignment but with a slightly lower frequency peak and wider distribution, which might be due to the errors in segmenting the mounting loop, because the absorption coefficient of the mounting loop can have a 10-20% difference from those of crystal and the mother liquor.

## S2 USAGE OF NANO BANANA

### S2.1 WORKFLOW

1. Start a new session with Nano Banana.

2. Upload all six images at once. Remember the order: Images 1–5 are the style references, and Image 6 is the content source.

3. Send **Prompt 1**.

4. The model will respond with a text summary of the style; it will not generate an image yet. Review this summary and correct it if needed.

5. Once the context are satisfied, send **Prompt 2** in the same chat session. The model will use its memory of the style profile from the previous step to generate the final image.

### PROMPT 1: THE ANALYSIS & SYNTHESIS PROMPT

This prompt instructs the AI to analyze Images 1–5 and create a detailed, written "style profile."

**Prompt.**

> You are an expert in scientific tomography projection image analysis. I have uploaded six images.
> **Images 1 through 5** are real examples of raw synchrotron X-ray projections.
> **Image 6** is a clean content image that we will use later. This is a simulated projection image from a manually segmented 3D volume by taking the Beer-Lambert law voxel-wise along a single axis.
>
> Your first task is to analyze Images 1–5 only. Do not modify any images yet.
>
> Please study the five examples and generate a detailed, written "Style Profile" that describes their shared visual characteristics. Organize your analysis under the following headings:
>
> 1. **Tonal Profile**: Describe the typical histogram, brightness, contrast, and black-level offset.
> 2. **Noise Signature**: Describe the combination of fine-grained (shot) and underlying (electronic) noise.
> 3. **Sharpness & Edge Quality**: Describe the characteristic blur or softness of the features (MTF roll-off), as well as the edge effect due to phase contrast.
> 4. **Common Artifacts**: Describe the typical appearance, intensity, and frequency of ring artifacts, streaking, and any large-scale background variations like cupping.
>
> Your output for this step must be text only. Do not generate an image.

### PROMPT 2: THE APPLICATION & GENERATION PROMPT

After the model provides the text summary, we send this second prompt. It leverages the model's short-term memory of the style profile it just created.

**Prompt.**

> Excellent, that is a perfect description of the style.
> Now, for the second and final step:
> Please apply the exact "Style Profile" you just described to Image 6.
> Remember to preserve the geometry and structure of Image 6 perfectly. The final output should be a new, realistic synchrotron projection that strictly adheres to the characteristics you outlined.
> Please export the final image in 16-bit grayscale. No watermark.

## S3 EXPERIMENTAL DETAILS

**Training Data** There are a total of 100 training/validation datasets, comprising 10 real experimental datasets, 90 simulated datasets, and 5 test datasets of real samples. The 100 datasets are split into 80 training datasets and 20 validation datasets for cross-validation (Isensee et al., 2021). Each real dataset includes a paired unlabelled 3D tomography reconstruction of crystal samples along with their manual segmentation. The unlabelled simulated datasets are created using FBP tomography reconstruction from 900 simulated tomography projection images based on an annotated 3D model, as illustrated in Figure 3.

**Hyper-parameters of training** We trained nnU-Net v2.4.1 (commit 9945333) with dynamic-network-architectures 0.4.2, batchgenerators 0.25.1, and PyTorch 2.8.0. The system ran on an AMD Instinct MI300X GPU (ROCm 6.2.6). To make runs repeatable, we fixed the Python, NumPy, and PyTorch random seeds and used deterministic settings (deterministic = true, benchmark = false). Inputs were resampled to 1.0×1.0 mm in 2D and 1.0×1.0×1.0 mm in 3D using cubic interpolation for images and nearest-neighbor for labels, then normalized with nnU-Net's CTNormalization (intensities clipped to [28, 243] and z-scored per scan). We did not apply data augmentation. The model (batch size 3) was trained for 1000 epochs with 250 iterations per epoch using SGD (initial LR 0.01, momentum 0.99, Nesterov is on, weight decay $3 \times 10^{-5}$), a Dice+Cross-Entropy loss with deep supervision, and a PolynomialLR scheduler (power p = 0.9, decaying the LR to zero over the full training). Checkpoints were written every 50 epochs, and we selected the checkpoint with the best validation Dice on the best fold for test inference. At test time we used sliding-window tiling (1024×1024 tiles, 0.5 overlap), mirroring along the two in-plane axes, and test-time augmentation [on/off]. Post-processing retained the largest connected component per class. We report [Dice/IoU], averaged per case.

## S4 SCIENTIFIC RESULTS OF THE CRYSTALLOGRAPHY EXPERIMENTS

| Metric | Test Sample 1 | | Test Sample 2 | |
|---|---|---|---|---|
| | Manual | Prediction | Manual | Prediction |
| Completeness (%) | 96.5 | 96.5 | 100.0 | 100.0 |
| Multiplicity | 21.3 | 21.3 | 13.7 | 13.7 |
| I/sigma | 25.5 | 23.1 | 26.5 | 22.7 |
| Rmerge(I) | 0.102 | 0.105 | 0.101 | 0.114 |
| Rmeas(I) | 0.105 | 0.107 | 0.104 | 0.118 |
| Rpim(I) | 0.021 | 0.021 | 0.026 | 0.030 |
| CC half | 0.998 | 0.998 | 0.997 | 0.995 |
| Anomalous completeness | 97.3 | 97.3 | 99.8 | 99.8 |
| Anomalous multiplicity | 11.7 | 11.7 | 7.7 | 7.7 |
| Anomalous correlation | 0.006 | -0.008 | 0.519 | 0.405 |
| Anomalous slope | 0.819 | 0.684 | 1.789 | 1.597 |
| dF/F | 0.079 | 0.080 | 0.070 | 0.074 |
| dI/s(dI) | 0.952 | 0.836 | 1.695 | 1.534 |

Table 3: Comparison of core metrics in crystallography experiments between Manual Segmentation and Prediction for Test Sample 1 and Test Sample 2.

### S4.1 EXPLANATION OF METRICS

The accuracy of the final molecular structure in crystallography depends heavily on these metrics. Completeness ensures that the dataset includes sufficient information to construct a reliable structure. Multiplicity, the number of observations for each reflection, helps reduce random errors, improving the robustness of the dataset. I/sigma measures the signal-to-noise ratio; higher values indicate clearer data, directly enhancing the precision of atomic positions. Rmerge, Rmeas, and Rpim evaluate the consistency of repeated measurements, with lower values indicating fewer errors and more accurate electron density maps. CC half assesses the internal consistency of the dataset, crucial for validating its quality. Metrics such as anomalous completeness, multiplicity, and corre-

lation measure the ability to capture subtle signals, essential for resolving features like chirality or metal centers. Finally, dF/F and dI/s(dI) indicate the strength and clarity of these signals, directly affecting the accuracy of fine structural details.

## S4.2 RESULTS ANALYSIS

For Test Sample 1, the predicted results closely align with the manual segmentation, as indicated by minimal differences across most metrics. Completeness, multiplicity, and CC half are identical, ensuring that the dataset remains reliable for accurate structural determination. The I/sigma metric shows a modest decrease of 9.4% (23.1 vs. 25.5), suggesting a slight increase in noise, though the impact on the precision of atomic positions is minimal. Precision metrics such as Rmerge (0.105 vs. 0.102) and Rmeas (0.107 vs. 0.105) exhibit small increases of 2.9% and 1.9%, respectively, indicating only minor reductions in reproducibility. These results align with the histogram of absorption factors as illustrated in Figure S1, which shows a high degree of overlap between manual and automatic segmentation. The frequency peaks and overall shapes of the histogram closely match, highlighting the model's strong performance in replicating the absorption factor distribution for all three materials.

For Test Sample 2, the differences between manual and predicted results are more pronounced. The I/sigma metric decreases by 14.3% (22.7 vs. 26.5), reflecting increased noise that could affect the precision of atomic positions. Precision metrics show larger discrepancies: Rmerge increases by 12.9% (0.114 vs. 0.101), and Rmeas rises by 13.5% (0.118 vs. 0.104), indicating reduced measurement consistency, which may blur the electron density map. The anomalous correlation decreases by 22.0% (0.405 vs. 0.519), and the anomalous slope drops by 10.7% (1.597 vs. 1.789), suggesting weaker detection and reduced consistency of anomalous signals. The histogram for absorption factors supports this observation, showing a slightly lower frequency peak and a wider distribution compared to Test Sample 1, as illustrated in Figure S1. Despite these challenges, the overall overlap in the histograms suggests that the automatic segmentation effectively captures the main trends in absorption factor distribution, providing a solid foundation for further refinement. This indicates that the current model already performs well in addressing complex datasets and shows promise for achieving even better results with targeted improvements.

