# OpenReview forum: "CrystalSeg: Automating Synchrotron Tomographic Reconstruction Segmentation for Crystallography with Physically Guided Simulations"
_ICLR.cc/2026/Conference — ICLR 2026 Conference Withdrawn Submission_

### Official Review · Reviewer_vowa · 2025-10-23

**Soundness:** 2
**Presentation:** 1
**Contribution:** 2
**Rating:** 2
**Confidence:** 3

**Summary:**

This paper proposes CrystalSeg, a simulation and segmentation pipeline for synchrotron X-ray tomography images. The pipeline can generate large-scale, high-quality datasets through physically derived simulations and image segmentation. However, the scope of experimental validation is limited, the model architecture lacks innovation, and the work does not provide open-source support for reproducibility. The simulation component relies purely on physics-based modeling, while the segmentation module is a straightforward nnU-Net implementation without task-specific tailoring. Overall, the contribution is more suitable for database and benchmark, but it remains insufficiently comprehensive to serve as a robust benchmark study.

**Strengths:**

The proposed physically driven simulation approach is feasible, and large-scale, high-quality data generation represents an effective way to enable AI-based spectral analysis to achieve real-world impact. According to the authors’ introduction, the proposed data simulation method appears to have merits in terms of both realism and computational efficiency.

**Weaknesses:**

The proposed model, CrystalSeg, mainly covers two components: data simulation and segmentation. However, only the segmentation part is directly related to ML model, and it merely adopts a standard nnU-Net, which limits the novelty of the work. In addition, the overall paper structure is not well organized, there is excessive contextual description of the physical simulation process (which could instead be placed in the appendix), while the model-related sections are insufficiently detailed. The paper does not clearly introduce the model architecture, data processing workflow, or training strategy.

**Questions:**

+ The simulated data should be made publicly available to enable validation and reproducibility.
+ The validation currently relies on only two real samples, which is insufficient to demonstrate generalization across different crystal morphologies, wavelengths, and noise conditions. It is recommended to:
  + Evaluate it on at least 5–10 additional samples and provide the statistical results.
  + Report segmentation and absorption-correction performance under varying energy ranges.
+ The segmentation study relies solely on nnU-Net. Please comparing with other representative 3D segmentation architectures, such as V-Net, TransUNet, Swin-UNETR, or DeepLabV3+.
+ Beyond SSIM and PSNR, please include quantitative evaluations of absorption spectra, phase-contrast profiles, and edge-enhancement fidelity to better assess the realism of the simulated data.

---

### Official Review · Reviewer_d2Em · 2025-10-28

**Soundness:** 2
**Presentation:** 3
**Contribution:** 2
**Rating:** 2
**Confidence:** 3

**Summary:**

The paper introduces CrystalSeg, a physics-guided, GPU-accelerated pipeline that (i) programmatically generates CAD-based 3D crystal/loop/liquor scenes, (ii) renders phase-contrast projections via multislice wave propagation under the Fresnel approximation with a pragmatic detector model (Gaussian PSF blur, Poisson shot noise, Gaussian read noise, and column-wise gain to induce ring artifacts), (iii) reconstructs volumes by FBP, and (iv) trains nnU-Net to segment crystal, mother liquor, and mounting loop. Training on Real+Syn outperforms OnlyReal and OnlySyn. Physics-guided simulation also matches real projections/reconstructions better than absorption-only and a diffusion “style-transfer” tool. The authors position this as enabling fully automated ray-tracing absorption correction, markedly reducing manual effort and addressing a practical bottleneck in long-wavelength crystallography.

**Strengths:**

- Long-wavelength crystallography needs accurate per-voxel segmentation for absorption correction, manual work is a bottleneck. The work is well-scoped to this pain point and ties to validation.
· Physically grounded simulator that models phase contrast, δ/β materials, detector response, and common artifacts, yielding more realistic projections/reconstructions than absorption-only surrogates.
· Physics-guided projections/recons exceed absorption-only in SSIM/PSNR for two samples, and synthetic data substantially boosts segmentation when mixed with real data.
· Include task-level validation, demonstrated downstream utility for automated absorption correction, indicating impact beyond proxy segmentation metrics.

**Weaknesses:**

· Training uses synthetic datasets with only 5 real test sets. Beamlines, energy and sample-morphology diversity are not systematically evaluated. Generalization beyond the reported setting is unclear.
· Results hinge on nnU-Net only, no comparisons to other strong 3D segmentation baselines, nor to real-only + heavy augmentation / self-training controls.
· There is no ablation study for phase-contrast, PSF/noise, or ring-artifact modeling. The “Nano Banana” comparison is illustrative but not a physics baseline, absorption-only is the only true ablation.
· The contribution asserts the “first fully automated” solution for absorption correction; evidence is compelling for two case studies but lacks tests across varied samples/beamlines.
· Downstream crystallographic validation is limited, only two case studies are shown.
· The paper states variation in refractive indices and randomization, but the sampled ranges and their match to beamline conditions are not tabulated. Ring-artifact modeling via column gain may not capture diverse ring etiologies.
· The manuscript claims hours to seconds reduction, but does not report generation/training/inference timings on stated hardware.
· Minor wording issue, contributions list says “ray-racing” instead of “ray-tracing.”

**Questions:**

· Generalization: How does performance vary across beamlines/energies, detector PSFs, and significantly different crystal morphologies? Any cross-site test sets?
· Synthetic–real gap: Do you measure distribution shift between synthetic and real reconstructions (e.g., edge statistics or learned feature distance), and how does it correlate with segmentation error?
· Simulator ablation: Can you quantify segmentation gains from (i) phase contrast vs. absorption-only, (ii) PSF/noise, and (iii) ring-artifact modeling?
· Baselines: How does Real+Syn compare with (i) Real-only + heavy augmentation and Real-only + self-training to isolate the marginal benefit of simulation, and (ii) competitive 3D segmentation baselines.

---

### Official Review · Reviewer_7VR2 · 2025-11-01

**Soundness:** 2
**Presentation:** 3
**Contribution:** 2
**Rating:** 4
**Confidence:** 4

**Summary:**

The paper proposes a simulation method for long-wave length X-ray crystallography. The simulation method is guided by the physical model of the crystallography technique. As a downstream task, the paper uses segmentation where training a segmentation network on their simulated data shows better performance than training it only on real data.

**Strengths:**

+ The paper is overall well written and organized
+ The experiments, despite being not extensive, showcases the importance of their simulation method
+ The paper addresses an important application problem in computational chemistry/biology domain

**Weaknesses:**

- The number of baselines is very limited and given the Figure 4, Nano Banana does not seem like a strong baseline. I understand that niche domains often have few baselines, but given the nature of the problem, GAN-based unpaired image translation frameworks could have been also used. text-to-3D frameworks can also be tested just to show how those general-domain methods would work in this niche domain.

- There are alternate solutions for limited annotated data for segmentation as well. There are methods for unsupervised segmentation, weakly-supervised segmentation, few-shot segmentation and many more. The paper uses simulation to create annotated data for supervised training without exploring how the alternative approaches would work. This undermines the central motivation of the work.

- Failure cases are not analyzed for simulation or segmentation.

**Questions:**

How does GAN-based unpaired image translation methods perform for 3D simulation?
How does the semi-supervised segmentation method perform when you use labels for simulated data and no labels for real data?
How does unsupervised segmentation perform on real-data?

**Details Of Ethics Concerns:**

The method may result in fake crystallography data of protein sequences, which can have ethical consequences. The paper did not discuss this.

---

### Note · Authors · 2025-11-29

I have read and agree with the venue's withdrawal policy on behalf of myself and my co-authors.